# Cell morphology and nucleoid dynamics in dividing *Deinococcus radiodurans*

Kevin Floc'h[1,4], Françoise Lacroix[1,4], Pascale Servant[2], Yung-Sing Wong [3], Jean-Philippe Kleman [1], Dominique Bourgeois [1] & Joanna Timmins [1]

Our knowledge of bacterial nucleoids originates mostly from studies of rod- or crescent-shaped bacteria. Here we reveal that *Deinococcus radiodurans*, a relatively large spherical bacterium with a multipartite genome, constitutes a valuable system for the study of the nucleoid in cocci. Using advanced microscopy, we show that *D. radiodurans* undergoes coordinated morphological changes at both the cellular and nucleoid level as it progresses through its cell cycle. The nucleoid is highly condensed, but also surprisingly dynamic, adopting multiple configurations and presenting an unusual arrangement in which *oriC* loci are radially distributed around clustered *ter* sites maintained at the cell centre. Single-particle tracking and fluorescence recovery after photobleaching studies of the histone-like HU protein suggest that its loose binding to DNA may contribute to this remarkable plasticity. These findings demonstrate that nucleoid organization is complex and tightly coupled to cell cycle progression in this organism.

[1] Univ. Grenoble Alpes, CEA, CNRS, IBS, F-38000 Grenoble, France. [2] Institute for Integrative Biology of the Cell (I2BC), CEA, CNRS, Univ. Paris-Sud, Université Paris-Saclay, 91190 Gif-sur-Yvette, France. [3] Univ. Grenoble Alpes, CNRS, DPM, 38000 Grenoble, France. [4] These authors contributed equally: Kevin Floc'h, Françoise Lacroix. Correspondence and requests for materials should be addressed to J.T. (email: Joanna.timmins@ibs.fr)

In all organisms, genomic DNA is compacted several orders of magnitude and yet must remain accessible for essential DNA-related processes including DNA replication, repair and transcription. In bacteria, packaging of genomic DNA is achieved by several mechanisms, including DNA supercoiling[1] and DNA compaction by nucleoid-associated proteins (NAPs), such as HU[2–4]. However, recent studies now indicate that additional factors, such as molecular crowding and depletion forces, also play important roles in determining the volume of the cell occupied by the nucleoid[5–7], and suggest that cell shape and size may critically influence nucleoid organization[7,8].

So far, a vast majority of the studies of bacterial nucleoids have focused on three model bacteria, *Escherichia coli*, *Bacillus subtilis* and *Caulobacter crescentus*, all of which are either rod- or crescent-shaped bacteria. Recent developments in single-molecule and genome-wide analytical techniques have started to shed light on how these model bacteria organize, compact and segregate their chromosomes, three processes that are intimately connected. Genetic, microscopy and chromosome conformation capture studies have revealed that nucleoids are organized into micro- and macrodomains[9], the origins of which are still unclear (reviewed in ref. [10]) and have been shown to adopt an overall helical arrangement[11–13], which is distributed along the longitudinal axis filling a large fraction of the cell volume ( ~ 70% in *E. coli*). Mapping of specific chromosome loci such as the origin (*oriC*) or the termination sites (*ter*) have also revealed that chromosomes in model bacteria can be seen to adopt two stereotypical configurations. Under certain conditions, they may arrange longitudinally, along an ori-ter axis with the *oriC* and *ter* regions located at opposite poles of the cell, or transversally in which both the *oriC* and *ter* are localized at mid-cell, and the left and right arms are located on either sides[2,13–15].

Although in all bacteria irrespective of their shapes, chromosome organization and segregation are tightly coupled to the division process, in cocci, the directionality of chromosome segregation has additionally been proposed to play a key role in positioning the division site[16]. The two most studied cocci are the ovoid *Streptococcus pneumoniae* and the spherical *Staphylococcus aureus*, two major human pathogens. So far, however, the small size of these cocci has largely restricted the study of nucleoid rearrangements and concomitant morphological changes occurring during their cell cycle. In contrast, *Deinococcus radiodurans* is a non-pathogenic, relatively large ( ~ 2 μm in diameter) coccus, displaying an outstanding resistance to DNA-damaging agents including ionizing radiation, UV light and desiccation[17–19]. Several factors have been proposed to contribute to this outstanding phenotype: (i) a highly efficient and redundant DNA repair machinery, (ii) the presence of numerous antioxidant metabolites[18,20–22] that contributes to an increased protection of the proteome, and (iii) the unusual properties of its genome. *D. radiodurans* has indeed been shown to possess a complex, multipartite genome composed of four replicons each of which are present in multiple copies ranging from four to ten, depending on its growth phase[23–25]. Moreover, *D. radiodurans*' genome has been reported to be more condensed[26] than that of radio-sensitive bacteria, such as *E. coli*, and to adopt an unusual ring-like structure[27,28]. These unusual features have been suggested to facilitate genome maintenance and repair[29,30].

Using both spinning-disk time-lapse microscopy and super-resolution imaging, we have performed a detailed analysis of the morphological changes that occur at the cellular and nucleoid level, as *D. radiodurans* grows and divides in alternating perpendicular planes. These data reveal that *D. radiodurans* nucleoids are indeed highly condensed, while remaining surprisingly dynamic, adopting multiple distinct configurations as the bacterium progresses through its cell cycle. Studies of the dynamics of the highly abundant histone-like HU protein, which is the major NAP associated with genomic DNA in *D. radiodurans*[31] reveal that it only binds loosely to DNA, which may facilitate the structural rearrangements of the nucleoid. Finally, we followed the choreography of the *oriC* and *ter* loci of chromosome 1 during the various stages of the cell cycle and show that they exhibit very different distributions within the cell with the *oriC* loci being radially distributed around the centrally located *ter* sites. Taken together, these findings demonstrate that the properties of *D. radiodurans* make it particularly well suited for the study of nucleoid organization in cocci and provide new, compelling evidence, indicating that bacterial nucleoids are complex and dynamic entities that are tightly coupled to cell shape, cell cycle progression and septal growth.

## Results

**Morphological changes during *D. radiodurans* cell cycle**. To follow the morphological changes occurring during the growth of *D. radiodurans*, cells were labelled with the membrane dye Nile Red and deposited on an agarose pad for either time-lapse imaging on a spinning-disk confocal microscope or super-resolution imaging using Point Accumulation for Imaging in Nanoscale Topography (PAINT) microscopy (see Methods and Supplementary Fig. 1 for a comparison of the two imaging systems). All cell measurements presented below were extracted from super-resolution images (Fig. 1a and Supplementary Fig. 2), whereas the order and duration of the various stages of the cell cycle were derived from spinning-disk microscopy images and videos.

Exponentially growing *D. radiodurans* cells cultured in rich medium form diads[32] that transiently form tetrads before the start of a new cell cycle (Fig. 1b). *D. radiodurans* cells divide sequentially in two perpendicular planes[33,34]. Starting from an elliptical and largely symmetric diad (Phase 1), the cells initiate their cell cycle by a phase of cell growth (Phase 2) that leads to a slight invagination at the junctions between the septum originating from the previous cell division ($S_{-1}$) and the cell periphery. Phase 2 is also characterized by a significant decrease in the ellipticity of the cells (Fig. 1b and Supplementary Figs. 2 and 3). Phase 3 starts with the appearance of bright foci corresponding to the onset of the growth of the new septa ($S_0$). These foci appear in the centre of the $S_{-1}$ septum and in the middle of the opposite peripheral cell wall. We observed that one foci may appear before the other in a given phase 3 cell and septal growth in the two cells forming the diad were frequently seen to be asynchronous (e.g., cells with asterisk in Fig. 1a). Phase 4 corresponds to cells in which the growth of the new $S_0$ septa is partial ( sum of lengths of interior and exterior septa < 80% of *P*, the distance between the central septum and the opposite side of the cell; Supplementary Fig. 2) and in Phase 5 the $S_0$ septa are almost closed (sum of lengths of interior and exterior septa > 80% of *P*; Supplementary Fig. 2). Finally, in Phase 6 septal growth is complete and tetrads are formed (Fig. 1b). As the cells progress through Phases 3 to 6, the invagination at the diad junction becomes more pronounced and the length of the $S_{-1}$ septum progressively decreases (Supplementary Fig. 3) until disappearing completely at the end of Phase 6, as the tetrads fall apart to form two diads (Fig. 1b, c). In stationary phase cells, the tetrads no longer separate into diads and the number of Phase 6 cells increases markedly (Supplementary Fig. 4). Moreover, a substantial fraction of these tetrads engages in a new division cycle, but cell cycle progression is very slow and only a small number of octads are seen (Supplementary Fig. 4).

Various cell parameters were extracted from our images (Supplementary Figs. 2 and 3) and were used to determine the average shape, size and volume of cells at each phase of the cell

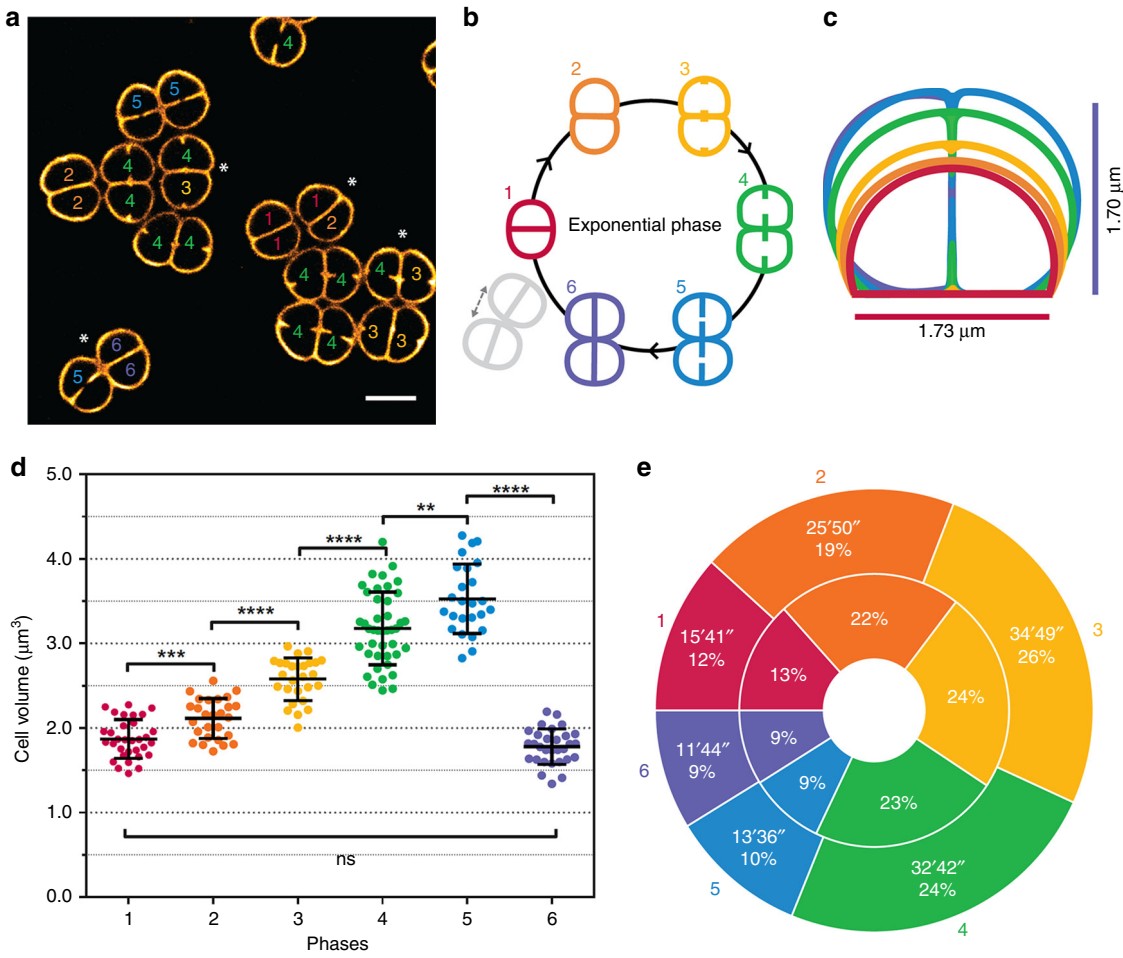

**Fig. 1** Morphological changes occurring during the growth of *D. radiodurans*. **a** PAINT imaging of live *D. radiodurans* cells stained with the membrane dye Nile Red. Numbers inside each cell correspond to their growth phase shown schematically in **b**. Diads marked with an asterisk are composed of two asynchronized cells. Scale bar: 2 μm. **b** Schematic representation of the different phases of the *D. radiodurans* cell cycle in exponentially growing bacteria. **c** On-scale representation of the average cell sizes and morphologies at the different phases of the cell cycle coloured as in **b**. Phase 6 cells are very similar to Phase 1 cells. **d** Changes in cell volume during the cell cycle ($N = 185$ cells, $N > 25$ for each phase). Data are represented as mean ± SD. Individual values are shown as dots. Cell volumes were calculated by measuring the cell parameters on PAINT images of exponentially growing wild-type *D. radiodurans* stained with Nile Red. The mean cell volume progressively increases from a volume of $1.87 \pm 0.23$ μm³ in Phase 1 to a volume of $3.53 \pm 0.41$ μm³ in Phase 5 just before cytokinesis. (***$P < 0.001$, ****$P < 0.0001$, ns: not significant; statistical test: non-parametric Mann–Whitney). **e** Duration of each growth phase in exponentially growing *D. radiodurans* cells. Interior annulus: distribution of exponentially growing cells in the different phases of the cell cycle when observed at a single time point ($N = 653$). Exterior annulus: duration of phases measured on exponentially growing cells observed by time-lapse imaging ($N = 44$). Data were collected from two independent experiments. Source data are provided as a Source Data file

cycle (Fig. 1c, d). *D. radiodurans* cells were found to grow linearly throughout their cell cycle (Supplementary Fig. 5), as seen by the progressive increase in cell volume starting from a mean volume of $1.87 \pm 0.23$ μm³ in Phase 1 until a volume of $3.53 \pm 0.41$ μm³ in Phase 5 just before cytokinesis (Fig. 1d). The change in cell volume was also accompanied by a change in ellipticity of the cells as they progress through the cell cycle (Supplementary Fig. 3). Starting from hemispheres in Phase 1, individual cells become quasi ellipsoids by the end of Phase 5. In contrast, in stationary phase, cell growth arrests and the mean volume of cells in tetrads decreases to $1.62 \pm 0.29$ μm³ (Supplementary Fig. 4).

To assess the duration of each growth phase in exponential cells, two approaches were used. Durations were either deduced from the distribution of phases in the population of exponentially growing cells when deposited on an agarose pad and directly observed under the microscope (the fraction of cells in each phase reflects the relative duration of these phases; Fig. 1e interior annulus) or by time-lapse imaging of cells (Fig. 1e exterior annulus and Supplementary Movie 1). Very similar results were

obtained for the two approaches and the total cell cycle duration measured in our time-lapse experiments ($134 \pm 11$ min) was close to the doubling time of 130 min derived from our growth curves performed on *D. radiodurans* cells grown in liquid medium in a shaking incubator (Supplementary Fig. 6). Cells spend 91% of their time as diads and 9% as tetrads, and septal growth (Phases 3, 4 and 5) occurs during more than half of the cell cycle. Phases 1, 5 and 6 are the shortest, lasting between 11 and 15 min, whereas Phases 2, 3 and 4 are longer and last 25 to 35 min (Fig. 1e).

**Cell growth occurs in both peripheral and septal cell walls.** In bacteria, cell growth is associated with expansion of the cell wall and peptidoglycan (PG) synthesis and remodelling. We thus investigated whether growth of the cell wall occurs (i) in the septal regions only, (ii) throughout the septal and peripheral cell walls, or (iii) only in the peripheral cell walls of *D. radiodurans* (Fig. 2a). For this, cells were pulse labelled with two dyes: BODIPY-FL 3-amino-D-alanine (BADA), a green dye that is stably incorporated into the pentapeptide chain during PG

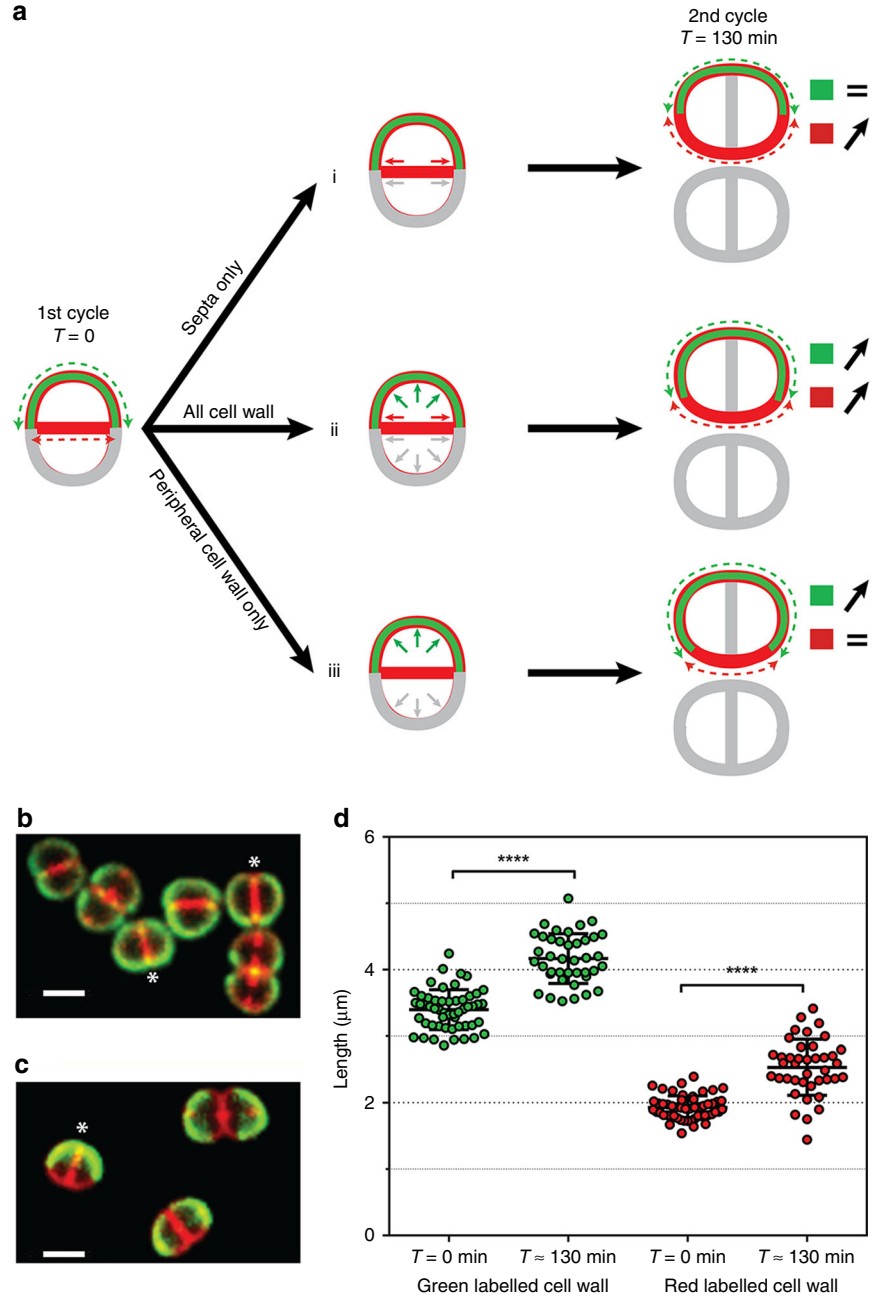

**Fig. 2** Cell growth occurs in both peripheral and septal cell walls. **a** Schematic representation of three possible models of cell wall growth for a cell initially in Phase 1 and undergoing a complete cell cycle. At $T = 0$ min, only the peripheral cell wall of the mother cell is stably labelled with BADA (green); the whole cell wall (peripheral and septal) is labelled with Nile Red (red). Nile Red signal suffers from a poorer signal/noise ratio than BADA; thus, the superposition of both colours does not appear yellow. Considering only the periphery of the new diad after a complete cell cycle ($T = 130$ min), if cell wall growth occurred (i) in the septal regions only, the length of the BADA-labelled cell wall would remain the same, whereas the length of the cell wall labelled only with Nile Red would increase; (ii) throughout the bacterial cell walls, both the length of the BADA-labelled cell wall and the cell wall labelled only with Nile Red would increase; (iii) only in peripheral cell walls, the length of the BADA-labelled cell wall would increase, whereas the length of the cell wall labelled only with Nile Red would remain the same. At $T = 130$ min, measurements were made on the peripheral cell walls only, which were originally labelled at $T = 0$. The newly grown septal regions at $T = 130$ min (gray in **a**, but stained with Nile Red in **c**) were not taken into account in these analyses. **b**, **c** Examples of *D. radiodurans* cells at $T = 0$ (**b**) and $T = 130$ min (**c**). Cells marked with an asterisk are examples of cells in Phase 1 that were used for the perimeter measurements and cell wall growth analyses (**d**). Scale bar: 2 μm. **d** Mean lengths of the green- and red-only-labelled cell walls at $T = 0$ min ($N = 53$) and $T = 130$ min ($N = 40$) for Phase 1 cells. Both significantly increase after a complete cell cycle. Data are represented as mean ± SD. Individual values are shown as dots. (****$P < 0.0001$; statistical test: non-parametric Mann–Whitney). Data were collected from three independent experiments. Source data are provided as a Source Data file

**Table 1 Description of bacterial strains used in this study**

| Strains | Description | Source or reference |
|---|---|---|
| *E. coli* BL21 (DE3) | *fhu*A2 *lon ompT gal* (λDE3) *dcm* Δ*hsdS* | Laboratory stock |
| *D. radiodurans* GY9613 | ATCC 13939, R1 | Laboratory stock |
| *D. radiodurans* GY15743 | *hbs::cherry*Ω*cat* | Laboratory stock |
| *D. radiodurans* GY15787 | *hbs::cherry*Ω*tet*, locus *ori::kan-par*Sc2$_{Bc}$/pFAP246 (P$_{spac}$-*parBc2Bc::drGFP, cat*) | 25 |
| *D. radiodurans* GY15800 | *hbs::cherry*Ω*tet*, locus *ter::kan-par*Sc2$_{Bc}$/pFAP246 (P$_{spac}$-*parBc2Bc::drGFP, cat*) | 25 |
| *D. radiodurans* GY17031 | *hbs::PAmcherry*Ω*kan* | 54 |

synthesis in the peripheral cell walls only and does not reorganize during the cell cycle, and Nile Red that diffuses into lipid membranes and thus labels the whole cell membrane, including in newly synthesized septa during the cell cycle. The samples were then washed to remove any unbound dye prior to imaging. The use of two lasers prevented *D. radiodurans* from growing under the microscope, so cells were observed shortly after the labelling procedure ($T = 0$) and then again after a complete cell cycle ($T = 130$ min). The mode of cell wall growth could be deduced from such images by comparing the lengths of green- and red-only-labelled cell walls per cell (Fig. 2a) at $T = 0$ with that observed after a complete cell cycle ($T = 130$ min). If growth is restricted to the septal region (model i), which is not labelled with the BADA dye at $T = 0$, the length of the green-labelled cell wall should remain constant after one cell cycle; if growth occurs throughout all cell walls (model ii), the lengths of both the green- and red-labelled cell walls should increase; and if growth is restricted to the outer periphery of the cells (model iii), then the red-only-labelled cell wall should remain constant (Fig. 2a). For ease of analysis, we performed our measurements on Phase 1 cells that displayed complete BADA (green) labelling of their peripheral cell wall, but only Nile Red staining of their S$_{-1}$ septum at $T = 0$ (Fig. 2b) and could thus be compared with our proposed models presented in Fig. 2a. After a complete cell cycle, Phase 1 cells clearly displayed an altered labelling pattern (Fig. 2b, c). Measurements revealed that the lengths of both green- and red-labelled regions had significantly increased (Fig. 2d), thus indicating that cell wall growth occurs throughout the entire cell surface of *D. radiodurans* cells including septal regions (Fig. 2a), as reported for *S. aureus*[35].

Next, we evaluated the position and angle of the newly growing S$_0$ septa relative to the previous S$_{-1}$ septa in exponentially growing cells. In all cells, septal growth was found to occur precisely in the middle (50% ± 1.5%) at a 90° ± 7° angle to the S$_{-1}$ septum (Supplementary Fig. 7). Early electron micrographs of *D. radiodurans* cells suggested that septal closure occurs through a closing door mechanism and not as a diaphragm[33]. Three-dimensional (3D) imaging of Nile Red-stained cells confirmed this (Supplementary Fig. 7). The closing septum grows from both sides of the cells leaving a gap that stretches across the whole cell.

**Nucleoid organization during the cell cycle.** To follow the 3D morphology of *D. radiodurans* nucleoids as a function of their cell cycle in exponentially growing and stationary phase bacteria, we used spinning-disk confocal microscopy on live cells. The nucleoids were visualized either by using a genetically modified strain of *D. radiodurans* in which the gene encoding the highly abundant HU protein was endogenously fused to mCherry (Table 1), or by staining the DNA with the Syto9 dye (Fig. 3). The high extent of colocalization observed when double staining *D. radiodurans* nucleoids with both HU-mCherry and Syto9 (Supplementary Fig. 8) confirmed that HU largely associates with genomic DNA. Importantly, the same nucleoid structures (described below), including the previously described ring-like or toroid structure, were observed irrespective of the mode of

labelling (Fig. 3 and Supplementary Fig. 8) and all four replicons of *D. radiodurans* assembled into a single nucleoid entity. When using Syto9, cell membranes could additionally be stained with Nile Red, to distinguish the different phases of the cell cycle (Fig. 3).

Nucleoid conformations were assessed in Syto9 and Nile Red-stained wild-type *D. radiodurans* cells and representative shapes of nucleoids were retrieved for each phase of the cell cycle (Fig. 3). In addition, time-lapse 3D imaging of either Syto9-stained nucleoids (Supplementary Movie 2) or HU-mCherry-expressing *D. radiodurans* (Supplementary Movie 3) was performed during a complete cell cycle to determine the chronological order in which these different conformations appeared. These videos clearly highlight the dynamic nature of the nucleoids. In Phases 1 and 2, a majority of nucleoids (respectively 52% and 42%) adopted toroidal shapes with an average diameter of 0.95 ± 0.10 µm (Supplementary Fig. 2). In addition, in Phase 1, a substantial fraction (28%) of nucleoids were found to form condensed, undefined structures, which were seen to be short-lived (minutes) in our time-lapses and rapidly became toroids, which appeared to be more stable and long-lasting. Due to the reduced resolution of the spinning-disk images in the Z-axis, we cannot rule out that these undefined structures may correspond to top views of toroids with an interior ring size below the resolution limit of the images. In Phase 2, 16% of nucleoids presented a square configuration with bright vertices and a significant fraction (21%) also adopted a more open configuration (open ring or crescent). In Phase 3, these two nucleoid configurations were found to be the most prevalent (respectively 25% and 30%) and the crescent-shaped nucleoids were positioned with their convex sides facing the S$_{-1}$ septum of the diad (Fig. 3a). The fraction of toroidal-shaped nucleoids decreased significantly in Phase 3 and instead they were seen to open up to form crescents. In some cases, an intermediary step was observed in which the toroids became squares with three or four of their vertices exhibiting strong fluorescent foci before opening up to adopt a crescent shape. In Phase 4 cells, most of these configurations disappeared and instead a large proportion of nucleoids were now elongated rods (54%) with an average length of 1.48 ± 0.25 µm and an apparent full width at half maximum of 0.52 ± 0.08 µm (Supplementary Fig. 2). More complex branched structures (35%) were also observed in Phase 4. Remarkably, these structures were all positioned with their long axis perpendicular to the future division axis (Fig. 3a). Time-lapse imaging revealed that these elongated structures resulted from the opening and stretching of the crescent-shaped nucleoids seen in Phase 3. In Phase 5 cells, in which the formation of the newly dividing septa was nearly complete, nucleoids appeared mostly as branched, elongated structures (35%), as seen in Phase 4, or as double rings connected by a thin thread (42%), in which each ring was positioned in one of the two future daughter cells. A clear constriction was observed around the centre of these nucleoids resulting from the two closing septal segments (Fig. 3a). Time-lapse imaging revealed that the rod-like structures predominantly found in Phase 4 progressively formed more complex, branched structures that eventually formed double crescents or rings

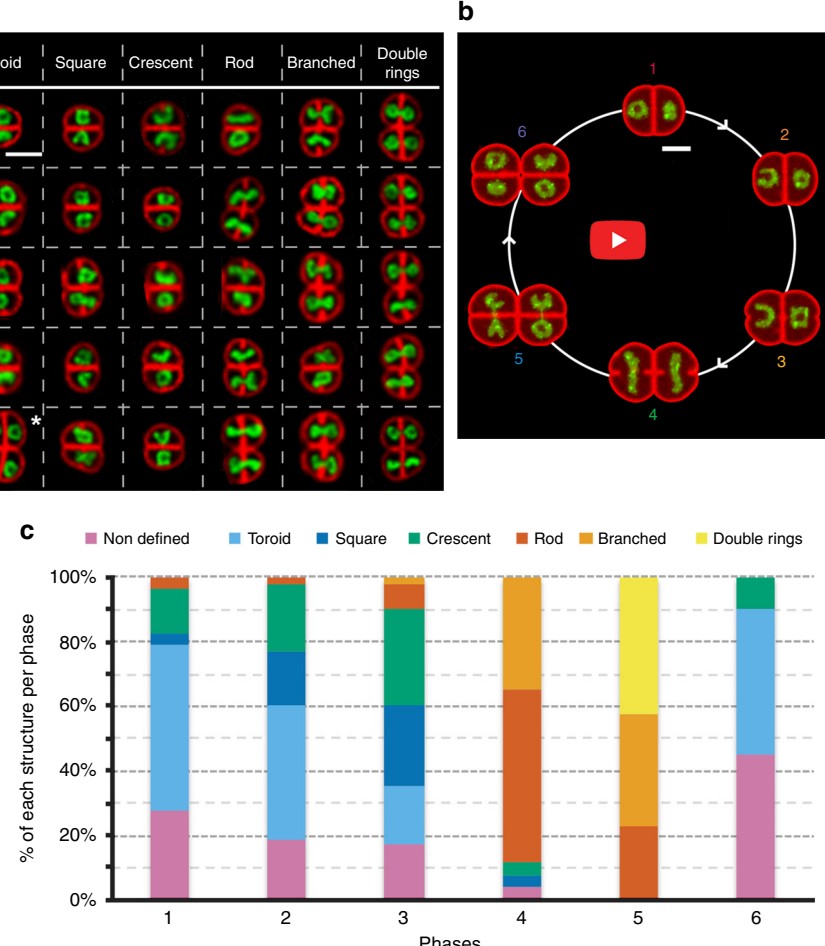

**Fig. 3** Nucleoid organization in exponentially growing *D. radiodurans*. **a** Array of the representative shapes of *D. radiodurans* nucleoids stained with the DNA dye Syto9. Cell membranes were stained with Nile Red. In each box, the representative shape is seen in the upper cell of the diad. In the case of tetrads, the representative shape is the cell marked with an asterisk. Scale: 2 μm. **b** Snapshot of the on-scale simulated reconstitution of the changes in cell shape and nucleoid structure occurring during the *D. radiodurans* cell cycle, illustrating the tight coordination of nucleoid rearrangements with septal growth (Supplementary Movie 4). Scale bar: 1 μm. **c** Distribution of the various nucleoid morphologies observed as a function of the phases of the cell cycle ($N = 232$ nucleoids; $N > 26$ for each phase). Source data are provided as a Source Data file

connected by short stretches of DNA. Finally, in Phase 6 cells, chromosome segregation was complete and the newly formed nucleoids presented similar configurations as those seen in Phase 1 cells, i.e., condensed, undefined structures (45%) or toroids (45%). These findings are summarized in a simulated movie that illustrates the principal morphological changes of the nucleoids observed in exponentially growing *D. radiodurans* cells as they progress through their cell cycle (Fig. 3b and Supplementary Movie 4). Interestingly, as observed previously when characterizing the different phases of the cell cycle, the two cells composing a diad were not necessarily synchronized in terms of nucleoid morphology. However, it was clear that the morphological changes at the nucleoid level were highly coordinated with cell cycle progression and septal growth (Fig. 3b).

**Nucleoid compaction.** Three-dimensional images of Syto9- and Nile Red-stained *D. radiodurans* were also used to determine the volume occupied by the nucleoid in *D. radiodurans* as a function of its cell cycle (Fig. 4a; Methods). For comparison, the same procedure was used for analysis of *E. coli* cells (Table 1 and Supplementary Fig. 9). Unlike the cell volume (Fig. 1c), the nucleoid volume did not increase linearly throughout the cell cycle (Fig. 4a). Instead, it was found to remain constant

($\sim 0.7 \ \mu m^3$) during Phases 1 to 3 and then to increase in Phases 4 and 5, to reach a maximum volume of $\sim 1.2 \ \mu m^3$ just before cytokinesis. The newly formed nucleoids in Phase 6 exhibited the smallest volumes ($\sim 0.6 \ \mu m^3$). As a result, the fraction of the cell occupied by the nucleoid varied as a function of the cell cycle from a minimal value of $0.30 \pm 0.06$ observed in Phase 4 to a maximal value of $0.36 \pm 0.06$ detected in Phase 1 cells (Fig. 4b). In comparison, the mean cell and nucleoid volumes of *E. coli* cells were found to be $1.33 \ \mu m^3$ and $0.83 \ \mu m^3$, respectively, corresponding to a cell fraction occupied by the nucleoid of 0.65 (Supplementary Fig. 7), in agreement with previously reported values[11]. These quantitative measurements thus confirm that *D. radiodurans* possesses a highly condensed nucleoid occupying approximately one-third of the cell volume, but also reveal that its level of compaction varies during its cell cycle. We also determined the mean volume of Syto9-stained nucleoids in *D. radiodurans* expressing HU-mCherry and found that these were slightly higher than that of wild-type cells (Supplementary Fig. 8), perhaps as a consequence of steric hindrance caused by the mCherry label, which may alter the DNA-binding properties of HU. Exponentially growing *D. radiodurans* cells have been reported to possess four to ten copies of their 3.2 Mbp genome[25], so the DNA density in an average *D. radiodurans* nucleoid ($\sim 0.7 \ \mu m^3$) should be in the range of 18–46 Mbp μm$^{-3}$. In

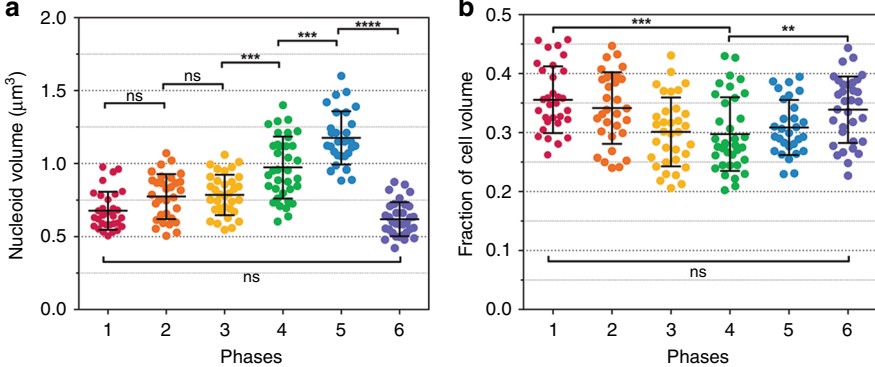

**Fig. 4** Nucleoid compaction as a function of *D. radiodurans* cell cycle. **a** Changes in nucleoid volume as cells progress through their cell cycle. Nucleoid volumes were retrieved from the three-dimensional signal of Syto9-labelled nucleoids of live, exponentially growing *D. radiodurans* cells. The nucleoid volume is relatively constant during the first three phases (~ 0.7 µm³) and then progressively increases to reach a maximal volume of ~ 1.2 µm³ in Phase 5 cells just before cytokinesis. **b** The fraction of the cell volume occupied by the nucleoid was retrieved by dividing the nucleoid volume of individual cells by their matched cell volume, derived from spinning-disk images of dual-labelled Syto9- and Nile Red-stained cells, using the cell measurements presented in Supplementary Fig. 2. The fraction of the cell occupied by the nucleoid significantly decreases during the first three phases (from 0.35 to 0.30), and then remains around 0.30 until cytokinesis. After septal closure, the fraction increases again to 0.34. **a, b** Data are represented as mean ± SD ($N = 195$, $N > 29$ for each phase). Individual values are shown as dots. (*$P < 0.05$, **$P < 0.01$, ***$P < 0.001$, ****$P < 0.0001$, ns: not significant, statistical test: non-parametric Mann–Whitney). Source data are provided as a Source Data file

*E. coli*, which possesses one to two copies of a 4.6 Mbp genome, the DNA density in its nucleoid should be between 5.5 and 11 Mbp µm⁻³, corresponding to a density at least three times lower than that observed in *D. radiodurans* cells.

Finally, Syto9- and Nile Red-stained stationary phase *D. radiodurans* tetrads in Phase 6 were also analysed to determine the level of nucleoid compaction in such cells (Supplementary Fig. 4). The average fraction of the cell occupied by the nucleoid was found to be 0.19, which is significantly lower than the highest level of DNA compaction measured in exponentially growing Phase 4 cells (Fig. 4b).

**Nucleoid and HU dynamics**. To further characterize *D. radiodurans* nucleoids in exponentially growing bacteria, we used super-resolution photoactivated localization microscopy (PALM) to image a *D. radiodurans* strain in which the most abundant NAP, HU, was fused to PAmCherry (Table 1). PALM experiments were performed exclusively on live bacteria, because fixation protocols strongly affected the nucleoid organization (loss of diverse shapes and decondensation of the DNA). Several of the nucleoid morphologies described above (e.g., crescent- and rod-shaped nucleoids) could be seen in the reconstructed PALM images, but most of the distinguishing features and details were lost (Fig. 5a). Although PALM provides higher spatial resolution, acquiring a complete dataset in PALM microscopy takes several minutes on our set-up. The loss of details in our PALM images thus most likely reflects the dynamic nature of the nucleoids in the second-to-minute timescale, as confirmed by a spinning-disk microscopy time-lapse experiment in which images of HU-mCherry were acquired in live *D. radiodurans* with short interval times (Supplementary Movie 5). This time-lapse video reveals that even though most of the nucleoid conformations last for several minutes (Fig. 3 and Supplementary Movies 2 and 3), the nucleoids are mobile and can be seen to rotate and reorganize in the second timescale.

HU has been shown to play a major role in the compaction and organization of the nucleoid in *D. radiodurans*[25,31] and to be essential for viability[36]. We therefore wondered whether HU may contribute to the remarkable plasticity of *D. radiodurans* nucleoids. Both single-molecule and ensemble microscopy approaches were used to probe the dynamics of HU proteins in live, exponentially growing *D. radiodurans*. Single-particle-tracking PALM (sptPALM) experiments were performed on the HU-PAmCherry-expressing *D. radiodurans* strain in which the trajectories of individual HU-PAmCherry molecules were reconstituted (Fig. 5b). Cumulative probability distribution (CPD) analysis[37] of 1355 individual tracks extracted from several datasets revealed that only a single population of molecules could be distinguished, which displayed a confined diffusion (with a confinement radius of ~ 400 nm; Fig. 5b) with a mean apparent diffusion coefficient of ~ 0.32 µm² s⁻¹. This value, which is significantly lower than that obtained for freely diffusing fluorescent proteins in bacteria (6-9 µm² s⁻¹)[38] but also higher than that of immobile, DNA-binding proteins ($D^* < 0.2$ µm² s⁻¹ [39,40]), clearly indicates that HU is not tightly associated with the genomic DNA. Ensemble measurements of HU mobility probed by fluorescence recovery after photobleaching (FRAP) on HU-mCherry-expressing *D. radiodurans* cells confirmed these findings and revealed that 99% of HU molecules were mobile exhibiting a half-life of 1.03 s (Fig. 5c, d). These data are consistent with a FRAP study of the *E. coli* NAP, H-NS[38], and suggest that even though HU largely colocalizes with genomic DNA, its association with DNA is only transient, which may allow the nucleoid to rapidly reorganize as the cells progress through their cell cycle.

**Distribution of *oriC* and *ter* sites during the cell cycle**. To follow the choreography of the *oriC* and *ter* loci during the cell cycle of exponentially growing *D. radiodurans*, we used engineered *D. radiodurans* strains, GY15787 and GY15800, in which a heterologous ParABS system is used to fluorescently label respectively the *oriC* and *ter* loci of chromosome 1 (Table 1), allowing them to be detected as bright foci within the cells[25]. Three-colour 3D spinning-disk images were recorded on such cells, to visualize green fluorescent protein (GFP)-labelled *oriC* or *ter* loci, hydroxycoumarin-amino-D-alanine (HADA)-stained cell walls and HU-mCherry-labelled nucleoids (Fig. 6a, d). When analysing these images, we chose to restrict our analysis to the bright ParB-labelled foci that could unambiguously be discriminated from the background fluorescence. This most likely explains why a reduced number of foci were counted per cell compared with the original study[25].

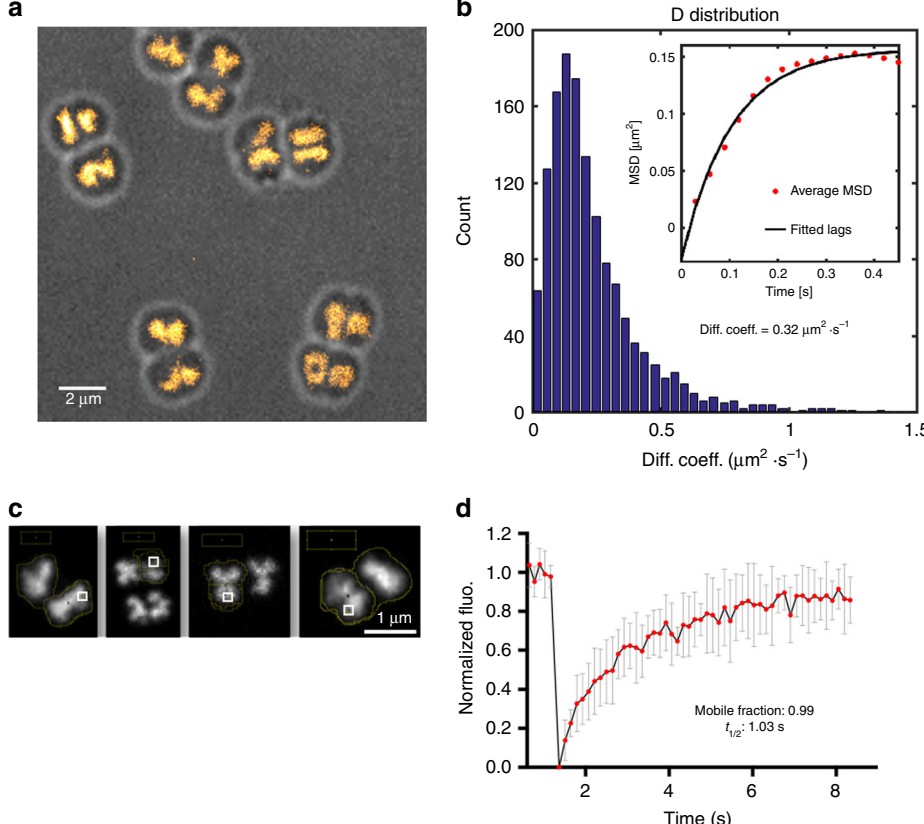

**Fig. 5** Nucleoid and HU dynamics in live *D. radiodurans*. **a** Reconstructed PALM image of HU-PAmCherry-expressing cells (stack of 15,000 frames with 50 ms frametime and acquired with constant 1 kW cm$^{-2}$ 561 nm laser and increasing 405 nm laser power). Several of the nucleoid morphologies illustrated in Fig. 3 can be seen in this image, but without a significative improvement in the perceived spatial resolution, due to the dynamics of nucleoids in live cells that change conformation in the second-to-minute timescale (Supplementary Movie 5). **b** Distribution of the apparent diffusion coefficients of 1355 single tracks of HU-PAmCherry measured by single-particle-tracking PALM in live, exponentially growing HU-PAmCherry-expressing *D. radiodurans*. Inset: mean square deviation (MSD) plot of the HU-PAmCherry tracks ($\Delta t = 30$ ms). The curve indicates that the molecules are confined with a confinement radius of 400 nm. The mean apparent diffusion coefficient derived from these 1355 tracks is of 0.32 µm$^2$/s. **c, d** Ensemble measurements of HU-mCherry mobility probed by fluorescence recovery after photobleaching (FRAP) on HU-mCherry-expressing live *D. radiodurans* cells. **c** Examples of HU-mCherry-labelled nucleoids used for FRAP experiments. The photobleached region (ROI1) is indicated with a white box. Scale bar: 1 µm. **d** Analysis of the recovery of the fluorescence signal of HU-mCherry after photobleaching. Fitting this full-scale normalized data reveals that 99 ± 1% of HU-mCherry molecules are mobile exhibiting a half-life of 1.03 ± 0.06 s. Presented data are the mean values ± SD (*N* = 10). Source data are provided as a Source Data file

Throughout the cell cycle, the number of *ter* foci in cells was systematically lower than that of *oriC* foci (Fig. 6a), as reported previously[25]. The number of *oriC* foci was found to double as the cells progress from Phase 1 to Phase 5, going from a mean number of 2.25 foci per cell to 4.44 foci per cell in Phase 5 (Fig. 6b). After septal closure, in Phase 6, the number of *oriC* foci was almost divided by 2, reaching 2.38 foci per cell. This suggests that DNA replication occurs all along the cell cycle until all copies of *oriC* from chromosome 1 have been duplicated. In contrast, the number of *ter* foci remains much lower around 1.09 foci per cell and constant for most of the cell cycle until Phase 5 where it increases to reach 1.38 foci per cell.

We also investigated the localization and distribution of the two loci within the cells (Fig. 6c). The *ter* foci were found to be specifically located to the central region of cells during most of the cell cycle. It was only in Phases 1 and 2 that the *ter* foci were found to be slightly off-centre. As the cells progress from Phase 1 to 3, the *ter* loci progressively move away from the S$_{-1}$ septum to relocate in the centre of the growing cells. In all phases, however, a majority of *ter* foci were located along the new division axis and 47% of them were still localized in this region at late stages of septal closure in Phase 5, just before cytokinesis, indicating a very late segregation of *ter*. In contrast, the *oriC* foci displayed a very different cellular

distribution and could be seen throughout the cell cytoplasm with only a minor fraction localized along the new division axis and in the centre of the cells. Multiple *oriC* foci were thus seen to be radially distributed around a single centrally located *ter* foci for most of the cell cycle. This was particularly visible in Phases 1 and 6. Interestingly, both *ter* and *oriC* seemed to be excluded from the peripheral region of the cell bordering the cell wall.

No clear correlation could be established between foci localization and nucleoid morphology. However, particular configurations could be seen (Fig. 6d). *ter* foci could, e.g., be observed occasionally in the centre of the toroid- and square-shaped nucleoids, i.e., in regions with low DNA density, devoid of HU-mCherry labelling. In the late stage of nucleoid segregation (Phase 5), *ter* foci were also seen to localize on the thin thread connecting the partially segregated nucleoids (double rings), close to the new division axis and the closing septum. *oriC*, in contrast, was often seen at the periphery of the nucleoids and occasionally at the outer tips of the branched nucleoids observed in Phase 4 cells.

## Discussion

In this work, we have characterized the cell cycles of exponentially and stationary growing *D. radiodurans*, enabling us to define

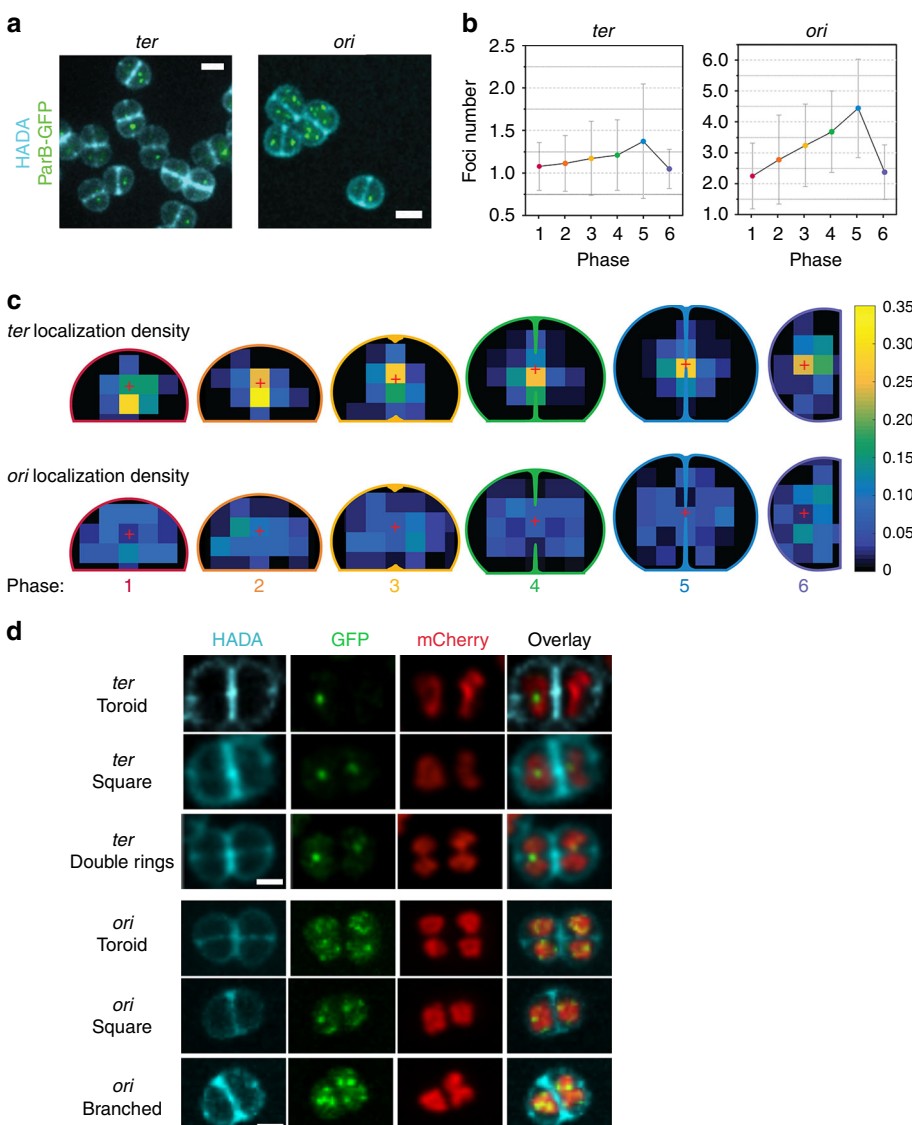

**Fig. 6** Distribution and choreography of *ter* and *oriC* loci of chromosome 1. **a** Images of *D. radiodurans* strains GY15787 and GY15800 expressing HU-mCherry and a heterologous ParB fused to GFP (green foci), with multiple copies of its cognate *parS* sequence inserted nearby chromosome 1 *ter* (left) or *oriC* (right) loci, respectively (Table 1), and stained with the cell wall HADA dye (blue). For clarity, the HU-mCherry signal is not included in these images. Scale bar: 2 μm. **b** Number of *ter* and *oriC* foci detected per cell at the different phases of the cell cycle retrieved from 3D images. Data are represented as mean ± SD. **c** Weighted distribution density of *ter* (top) and *oriC* (bottom) foci at the different phases of the cell cycle. The red cross indicates the centre of the cell. **b**, **c** *ter* ($N = 315$; $N > 37$ for each phase) or *oriC* ($N = 368$; $N > 36$ for each phase) foci were analysed and data were retrieved from two independent experiments. **d** Examples of *ter* and *oriC* foci localization relative to the different nucleoid conformations of *D. radiodurans*, expressing HU-mCherry (red) and ParB-GFP (green), and stained with the cell wall dye HADA (blue). Scale bar: 1 μm. Source data are provided as a Source Data file

eight distinct phases, and have followed the changes in the cell and nucleoid morphology occurring during these processes.

We determined that exponentially growing *D. radiodurans* diads grow throughout their cell cycle via remodelling of their entire cell surface (peripheral and septal cell walls) and divide in alternate orthogonal planes. As proposed in an early study of *D. radiodurans*[33], our 3D images confirmed that growth of the dividing septum occurs through a closing door mechanism and not as a diaphragm as has been reported for ovococci[41,42]. One side of the $S_0$ septum starts growing precisely from the middle of the $S_{-1}$ septum, whereas the other grows from the middle of the opposite peripheral cell wall, both at a 90° angle to the $S_{-1}$ septum. This observation is surprising and will be the subject of future studies. Septal growth lasts for two-thirds of the cell cycle (Phases 3–6). Asynchronization of the division process was frequently

observed: (i) between two cells that form a diad, resulting in cells of the same diad being in different phases of the cell cycle, or (ii) within a given cell, in which one side of the closing septum started growing earlier or faster than the other side, although, overall, no difference in length was observed between the two sides of the closing septum (Supplementary Fig. 3). In exponentially growing *D. radiodurans*, once the dividing $S_0$ septum closes (Phase 6), the newly formed tetrads are relatively short-lived, lasting for a dozen of minutes before splitting into two diads to initiate a new cell cycle (Phase 1). In Phase 6, cytokinesis results from the progressive increase in the curvature of the cell wall leading to a reduction in the length of the shared $S_{-1}$ septum, until the two diads eventually separate, creating a small gap between the two adjacent diads (see separating tetrad, bottom left in Fig. 1a). It was recently shown in *S. aureus* that the splitting of the diads into

single cells occurs on the millisecond timescale driven by the turgor pressure[35,43]. In *D. radiodurans*, visualizing this splitting process was more difficult, as the cells do not undergo any substantial morphological changes upon cytokinesis. Instead, the separation appears to be slower and progressive, most likely catalysed by the enzymatic processing of the cell wall in this region of the cell.

In every organism, nucleoid organization and cell division are intimately linked. The present study presents the first morphological characterization of the nucleoids of a coccus bacterium. The nucleoids of *D. radiodurans* were found to display a remarkable plasticity, adopting multiple, distinct conformations as the cells progress through their cell cycle. Our sptPALM and FRAP studies suggest that the dynamic nature of these nucleoids may be facilitated by the loose binding of the major *D. radiodurans* NAP, the histone-like protein HU, to the genomic DNA. To our knowledge, this is the first report of such a wide diversity of nucleoid structures in bacteria. In addition to the previously reported toroid shape, the nucleoid was also found to adopt an elongated branched structure that aligned parallel to the $S_{-1}$ septum. This arrangement perpendicular to the newly growing $S_0$ septum provides clear evidence that nucleoid organization and chromosome segregation are tightly coupled to the cell division process. In many model bacteria including *S. aureus*, *E. coli* and *B. subtilis*, nucleoid occlusion (NO) systems, typically composed of sequence-specific DNA-binding proteins, play a central role as spatial and temporal regulators of cell division by preventing the assembly of the divisome over the nucleoid[44]. Interestingly, in *D. radiodurans* Phase 3 cells in which septal growth is starting, three major conformations of nucleoids can be observed: toroids, squares and crescent-shaped nucleoids formed by the opening of the ring-like structures prior to their elongation and alignment with the $S_{-1}$ septum in Phase 4. Changes in the cell morphology and, in particular, start of septal growth thus seem to precede the major rearrangements of the nucleoid, which suggests that the nucleoid is not the major determinant of the division site in *D. radiodurans*. Although no NO system has so far been identified in *D. radiodurans*, we cannot exclude that such a system might also be at play in the regulation of cell division in this organism.

The Min system that prevents the formation of the Z-ring in aberrant locations[45] might be the main factor ensuring that cytokinesis occurs at mid-cell in *D. radiodurans*. Many spatial features have been proposed as key determinants of pole-to-pole oscillations of Min in *E. coli*, such as the highly negative membrane curvature at the poles[46], the longest axis of confinement[47], the longest possible distance for the diffusion of MinD[48] or the symmetry axes and scale of the cell shape[49]. *D. radiodurans* diads do not possess any poles per se. In Phase 1, cell ellipticity is close to 1 and cells possess an infinite number of symmetry planes (i.e., every plane orthogonal to the $S_{-1}$ septum). It is only when the cells start growing in Phase 2 (with a major change in the length of the major axis and no change in minor axis length leading to a decrease in the ellipticity of the cells) that the number of symmetry planes is reduced to two, both orthogonal to the $S_{-1}$ septum, one having its normal vector parallel to the long axis of the cell and the other with its normal vector parallel to the second short axis. A possible oscillation of MinCDE along the long axis of the ellipsoidal-shaped cell, parallel to the $S_{-1}$ septum, could be the basis for the precise placement and angle of growth of the dividing septum, orthogonal to the $S_{-1}$ septum and along the short axis of the cell. In such a scenario, the Min system would sense the geometry of the cell and start to oscillate in Phase 2.

As bacterial nucleoids fill a large fraction of the total cell volume, a common hypothesis suggests that the cell size and shape may play a major role in chromosome positioning and sizing. It was recently proposed that cell shape and size were 'sensed' indirectly by chromosomes via the pressure exerted by depletion forces induced by cytosolic crowders[8,50], which would promote the compact shape and central localization of nucleoids in cells. In the case of *D. radiodurans*, several studies have described its nucleoid as being highly condensed[26–30], but this visual impression is in part due to the relatively large size of *D. radiodurans* cells that have a mean volume of ~ 2.6 μm³. We therefore assessed the volume occupied by the nucleoids of *D. radiodurans* and compared them with that of *E. coli*, which had previously been reported to occupy ~ 70% of the cell volume[11]. Our measurements revealed that the mean volume of *D. radiodurans* nucleoids is very similar to that of *E. coli* (0.7 vs. 0.8 μm³); however, because of the larger cell volume, the fraction of the cell occupied by the nucleoid is significantly lower in the case of *D. radiodurans* and was found to vary from 0.30 to 0.35 depending on the phase of the cell cycle. This fraction even decreased below 0.2 in stationary phase cells, which are smaller and present highly condensed nucleoids. Such small cell fractions occupied by the nucleoid suggest either that depletion forces do not play a major role in nucleoid positioning and compaction in *D. radiodurans* or that *D. radiodurans* cells possess an unusually high concentration of crowders.

As *D. radiodurans* possesses multiple copies of its genome, compared with a single copy in *E. coli*, the average DNA density in *D. radiodurans* nucleoids is thus significantly higher than in *E. coli*, thereby confirming the impression that *D. radiodurans* nucleoids are particularly condensed. Interestingly, the nucleoids were found to be most compact in Phases 4 and 5, during the final stages of chromosome segregation. The specific positioning of the nucleoids in the centre of cells and parallel to the $S_{-1}$ septum, together with their increased levels of compaction during these late stages of the cell cycle, are reminiscent of metaphase chromosomes in eukaryotic cells that align at mid-cell prior to segregation of each one of the two chromosomes into the two daughter cells.

Finally, we compared the number and distribution of the *oriC* and *ter* loci of chromosome 1 as a function of the cell cycle. As previously reported[25], there were significantly more *oriC* foci in cells than *ter* sites irrespective of the stage of the cell cycle. This has been proposed to result from differences in the duration of periods of transient cohesion of the *oriC* and *ter* loci from the various copies of chromosome 1 after replication—this period being longer for *ter*—and/or a very late replication of *ter* compared with *oriC*[25]. Our cartography of the localizations of *oriC* and *ter* loci in *D. radiodurans* at each phase of the cell cycle also revealed very distinct distribution patterns for the two loci and allow us to propose a first model for nucleoid organization and choreography in a coccus bacterium possessing a complex, multicopy genome (Fig. 7). Chromosome arrangement in *D. radiodurans* is neither longitudinal nor transversal, but instead radial with *oriC* sites distributed all around centrally located *ter* sites. The *ter* loci were clearly retained and clustered by a yet unknown mechanism in the central region of the cells until just before cytokinesis (Fig. 7b, c). *ter* foci could indeed still be seen in the small gap formed by the closing septal doors in Phase 5 cells. Segregation of replicated *ter* loci occurred at a very late stage of the cell cycle (Phase 5; Fig. 7c) and the duplicated *ter* loci then progressively migrated to the centre of the two new sister cells. In contrast, *oriC* loci localized throughout the cell, with the exclusion from membrane proximal regions, ruling out any direct anchoring of *oriC* loci to the cell walls as has been observed in some bacteria[51–53]. Further work will be needed to understand the detailed choreography of this complex genome and how it relates to the various nucleoid structures observed in this study, but our findings suggest that cell shape may critically influence both nucleoid organization and chromosome arrangement.

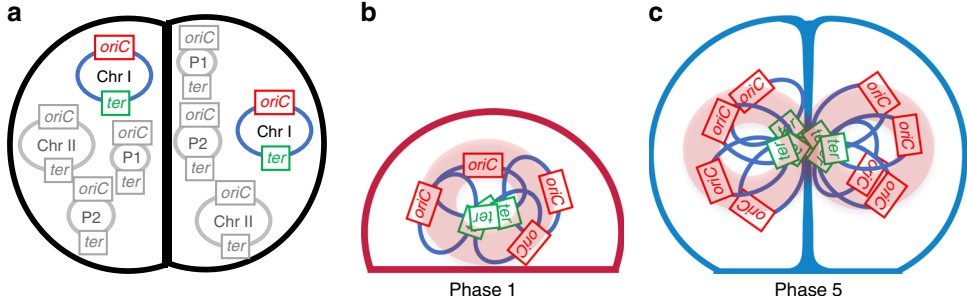

**Fig. 7** Radial arrangement and segregation of chromosome 1 in *D. radiodurans*. **a** *D. radiodurans* possesses multiple copies (4 to 10) of four replicons (2 chromosomes: Chromosome 1, Chr1, and chromosome 2, Chr 2, and two plasmids, P1 and P2), each of which possesses an origin of replication (*oriC*) and a termination site (*ter*). For Chr1, the *oriC* and *ter* loci are represented in red and in green, respectively. **b**, **c** Proposed arrangement of the multiple copies of Chr1 in Phase 1 and 5 cells. Typical nucleoid shapes observed in these two phases (Fig. 3c) are shown in pale pink. Chr1 is present in at least four copies in Phase 1 and this number doubles by the end of the cell cycle (Phase 5). The number of *ter* foci remains constant and close to 1 for most of the cell cycle, most likely as a result of the clustering of *ter* loci by an unknown mechanism in the centre of the cell. The *oriC* loci are radially distributed around the *ter* loci and are found throughout the cytoplasm with the exception of membrane proximal regions

## Methods

**Bacterial cultures**. All *D. radiodurans* strains (wild-type, HU-mCherry GY15743, HU-PAmCherry GY17031 and *oriC/ter*-labelled strains GY15787 and GY15800; Table 1) were grown at 30 °C with shaking in Tryptone-Glucos-Yeast extract 2x medium (TGY2x) supplemented with the appropriate antibiotics. *E. coli* strain BL21 (DE3) (Table 1) was grown at 37 °C with shaking in Luria Bertani medium. For microscopy experiments, *D. radiodurans* cells were pre-grown the day before and then diluted for an overnight growth until reaching exponential (OD$_{650}$ ~ 0.3-0.5) or stationary phase (OD$_{650}$ > 5). To establish the growth curve of wild-type *D. radiodurans* cells, cultures inoculated at different cell densities were measured every 2 h and then combined to obtain a complete growth curve. The final growth curve corresponding to the mean of three independent experiments was fitted to an exponential growth curve ($y = y_0 \ast \exp(K \ast x)$ with $y_0$ corresponding to the *y*-value at $T = 0$ and $K$ the growth rate) using GraphPad Prism 6 and the doubling time was calculated as $\ln2/K$.

**Sample preparation for microscopy**. Glass coverslides used for PALM/PAINT imaging of *D. radiodurans* were treated in an ozone oven for at least 10 min. Low-melting agarose (1.75%; Biorad) pads dissolved in minimal medium[54] (for PALM) or TGY2x (for spinning-disk microscopy) were poured on a cover slide inside a frame made with double-faced tape. A glass coverslip was placed on top of the agarose pad, to flatten the surface of the agarose while hardening. This coverslip was removed once the agarose had solidified. For imaging, bacterial cultures were centrifuged 3 min at 3000 × *g* and were resuspended in 10 µl of medium or washed in high-purity phosphate-buffered saline (GIBCO). One microlitre of this cell suspension was deposited on the pads. The drop was spread over the surface of the pad by rotating the coverslip. A second coverslip was placed over the agarose pad containing the sample, thereby immobilizing the bacteria on the pad.

To determine the growth phase and cell volumes, cells were incubated with Nile Red for 15 min at 30 °C, with agitation and then placed on an TGY agarose pad. For single time-point images acquired with the spinning-disk microscope, the Nile Red was used at a concentration of 30 µM. For correlative spinning-disk microscopy and PAINT imaging, the concentration was 300 nM. For PAINT imaging alone, the concentration was 15 nM. To determine nucleoid volumes, cells in exponential phase were incubated with Nile Red (30 µM) and Syto9 (150 nM) for 15 min at 30 °C, with agitation and then placed on a TGY2x agarose pad. In all cases, the cells were rinsed to remove excess dye prior to imaging.

For the PG labelling, cells in exponential growth phase were stained with the fluorescent BADA dye[55] at a concentration of 31 µM, incubated at 30 °C with agitation for 15 min. Then Nile Red was added (15 µM) for another 15 min at 30 °C with agitation. The cells were then rinsed in TGY2x at 30 °C, to remove unbound dyes. Cells were then deposited on TGY2x agarose pads for imaging ($T = 0$) or returned to the incubator until $T = 130$ min.

For time-lapse imaging of the cell growth and division process, cells in exponential growth phase were incubated with Nile Red (15 µM) for 15 min at 30 °C, with agitation. Cells were then rinsed in TGY2x before imaging. For time-lapse imaging of the nucleoids, exponentially growing HU-mCherry-expressing cells or Syto9-stained (150 nM for 5 min) wild-type cells were used. In all cases, cells were deposited on an TGY2x agarose pad designed with air holes to oxygenate the cells[56]. The samples were maintained at 30 °C during image acquisition.

To image *oriC*- or *ter*-labelled HU-mCherry-expressing *D. radiodurans* cells (Table 1), an overnight pre-culture was diluted 60× in TGY2x supplemented with 3.4 µg mL$^{-1}$ chloramphenicol and grown for a further 5 h until reaching an OD$_{650}$ between 0.3 and 0.5. Cells were then stained with HADA (30 µM) for 30 min and deposited on a TGY2x agarose pad.

**Image acquisition**. For PALM/PAINT imaging, wide-field illumination was achieved by focusing the laser beams to the back focal plane of a 100 × 1.49 numerical aperture (NA) oil-immersion apochromatic objective lens (Olympus). The intensity and time sequence of laser illumination at the sample were tuned by an acousto-optical tunable filter (AOTF; Quanta Tech). Near-circular polarization of the laser beams was ensured by inserting a polychromatic quarter-wave plate downstream the AOTF. Eight to 20 cells were imaged per field of view. Typically, for each sample, two or three fields of view on the same agarose pad were imaged and at least three independent experiments were performed on different days. Sample drift was corrected in ImageJ using gold nanobeads (Sigma) deposited on the agarose pads nearby the bacteria.

For single-particle-tracking experiments, data were acquired with continuous 561 nm light illumination, at low power (130 W cm$^{-2}$) using a frametime of 30 ms. The Trackmate plugin for ImageJ was used to localize the particles in each frame and to connect the coordinates into trajectories[57]. Simple Linear Assignment Problem Tracker generated the tracks, with a maximal allowed linking distance set to 200 nm and a maximal frame interval between two spots to be bridged set to 2. Only tracks that contained more than four points were exported into MATLAB for further processing. The TrackArt MATLAB software[37] was used to analyse trajectories and to determine CPDs and apparent diffusion coefficients. Trajectories shorter than eight frames or with a minimal individual mean square displacement fit $R^2$-values below 0.9 were filtered out.

Spinning-disk confocal microscopy was performed on the M4D cell imaging platform at IBS using an Olympus IX81 microscope and Andor iXon Ultra EMCCD Camera with the laser beams focused to the back focal plane of a 100 × 1.49 NA oil-immersion apochromatic objective lens (Olympus). The intensity and time sequence of laser illumination at the sample were tuned by an AOTF. For cell and nucleoid volume analyses, series of Z-planes were acquired every 100 nm using a piezo stage, whereas 200 nm was used for the multiplex imaging of *oriC*- and *ter*-labelled *D. radiodurans* strains. Fluorescence emission was collected through a quad-band Semrock™ Di01-T405/488/568/647 dichroic mirror and the corresponding single-band emission filters. For Nile Red time-lapse imaging (Supplementary Movie 1), single-plane image sets (two-dimensional (2D)) were acquired every 10 min, for a total period of 4 h, using very low 561 nm laser power (few micro-Watts) and 100 ms exposure times. Nucleoid time-lapse imaging of Syto9- or HU-mCherry-labelled cells (Supplementary Movies 2 and 3) was achieved by acquiring 3D images (Z-planes were acquired every 100 nm) every 5 (Supplementary Movie 3) or 10 min (Supplementary Movie 2) for a period of 3 h. An additional 3D time-lapse video of HU-mCherry was also performed with short interval times (20 s) over a period of 10 min (Supplementary Movie 5). HU-mCherry time-lapses were corrected for bleaching using histogram matching in ImageJ[58].

For FRAP measurements, a time-lapse acquisition of the fluorescence channel (mCherry) was performed for 60 timepoints at the fastest acquisition speed (10 timepoints before and 50 timepoints after photobleaching). Photobleaching (Andor FRAPPA module) was set at the wavelength used for imaging, targeting a single point (crosshair) of the analysed nucleoid. Bleaching was performed using a low-intensity setting (2% of the nominal maximum laser power), for 160 µs dwell time and 12 repeats, to achieve a sufficiently small bleached region and with a loss of at least 50% of the original signal intensity (mean signal loss of 60%).

**Data analysis**. To calculate the volume of each cell, an ellipse was fitted to the cellular membrane of Nile Red-labelled cells in 2D images (Supplementary Fig. 2a). Measurements of the shorter ($L_{short}$) and longer ($L_{long}$) axes of these ellipses were extracted from such fits. The distance between the central S$_{-1}$ septum and the extremity of the exterior of the ellipse along the short axis was measured ($P$), to

retrieve the distance between the septum and the centre of the ellipse along the short axis ($w$). With all these values, the total volume of the cell was computed using the equations presented in Supplementary Fig. 2a with the cells approximated as prolate ellipsoids. The length and width of the elongated and toroid-shaped nucleoids were measured as described in Supplementary Fig. 2c-d.

To determine the extent of BADA (green) vs. Nile Red (red) staining in Phase 1 cells, the perimeter of each cell was measured by fitting an ellipse to the Nile Red-labelled cell membrane. The fraction of cell wall labelled with BADA was then determined by measuring the two angles as shown in Supplementary Fig. 2b. The perimeter of Nile Red-only-stained membranes (without any BADA labelling) for an angle $\alpha$ is retrieved from the equation (1). The BADA-labelled perimeter was then determined by computing the lengths of the two sections of cell wall devoid of BADA labelling and subtracting them from the whole perimeter of the ellipse.

$$l = \int_0^\alpha \sqrt{\left(L_{\text{short}} * \cos(\theta)\right)^2 + \left(L_{\text{long}} * \sin(\theta)\right)^2} \, d\theta \qquad (1)$$

To determine the volume of nucleoids, Z-stacks of the Syto9 fluorescence signal corresponding to the nucleoids were deconvoluted in Volocity using seven iterative cycles and a calculated point spread function (PSF) (488 nm/520 nm). Cropped images in which the fluorescence signal was homogeneous throughout the full field of view were then exported to the Imaris (Bitplane) software, where 3D objects corresponding to the nucleoids were created based on the automatic thresholding of the fluorescence signal. The volumes of these objects were then extracted for further analysis. The same procedure and settings in Imaris were used to extract nucleoid volumes from all our samples, including E. coli nucleoids.

To evaluate the distribution of oriC and ter foci, Z-stacks of the GFP signal were iteratively deconvoluted using a calculated PSF (488 nm/520 nm) in Volocity. In the case of ter, Z-projections for each channel were concatenated into multi-channel stacks. In the case of oriC, the same multi-channel stacks were assembled, but we also verified that fluorescent foci were not lost by the Z-projection. Fluorescent foci were manually marked in ImageJ to extract their xy coordinates as well as the dimensions and phase of the cells in which they were found. The number of foci and their distribution were then automatically determined with respect to the average cell dimension of D. radiodurans cells at each phase of the cell cycle derived from PAINT imaging.

For FRAP data analysis, raw data corresponding to fluorescence intensity measurements from two regions of interest (ROI1: bleached area, ROI2: total nucleoid) were corrected for background fluorescence (ROI3: background) and then normalized using the full-scale method in EasyFRAP[59], and the resulting normalized data were fitted ($R^2 = 0.98$) to the following equation $y = M_{\text{frac}}*x/(t_{1/2} + x)$ in GraphPad Prism 6 in which $M_{\text{frac}}$ corresponds to the mobile fraction and $t_{1/2}$ to the half maximal recovery time of the mobile fraction.

The non-parametric Mann–Whitney statistical tests were performed with GraphPad Prism 6, to assess the difference in cell dimensions and nucleoid volumes during the cell cycle. Two-tailed P-values below 0.01 were considered as significant and were indicated with asterisks as follows: **$P \leq 0.01$, ***$P \leq 0.001$, ****$P \leq 0.0001$.

**Reporting summary**. Further information on research design is available in the Nature Research Reporting Summary linked to this article.

## Data availability

The source data underlying Figs. 1d-e, 2d, 3c, 4a-b, 5b, 5d, 6b-c and Supplementary Figs. 1d-e, 3a-c, 4b, 4d, 6, 7a, 8e and 9b-d are provided as a Source Data file. Other data that support the findings of this study are available from the corresponding author upon request.

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

## Acknowledgements

We thank Professor Yves Brun and Professor Michael Van Nieuwenhze for providing us with the BADA dye. This work was supported by CEA Radiobiology grant (A-IRBIO-01-19) and the CEA Irtélis programme (PhD grant of K.F.). This work used the M4D imaging platform of the Grenoble Instruct-ERIC Center (ISBG: UMS 3518 CNRS-CEA-UGA-EMBL) within the Grenoble Partnership for Structural Biology (PSB), supported by FRISBI (ANR-10-INBS-05-02) and GRAL (ANR-10-LABX-49-01), financed within the University Grenoble Alpes graduate school (Ecoles Universitaires de Recherche) CBH-EUR-GS (ANR-17-EURE-0003). IBS acknowledges integration into the Inter-disciplinary Research Institute of Grenoble (IRIG, CEA).

## Author contributions

K.F. and J.T. designed the research. K.F., F.L. and J.P.K. performed the microscopy experiments. P.S. provided the genetically modified strains of *D. radiodurans*. Y.S.W. synthesized the HADA dye. K.F., F.L., J.P.K., D.B. and J.T. analysed the data. K.F., D.B. and J.T. wrote the manuscript and all authors discussed the results and approved the manuscript.

## Additional information

**Competing interests:** The authors declare no competing interests.

