## [Peer Review File · Nature Communications]

Reviewers' comments:

Reviewer #1 (Remarks to the Author):

The authors do an impressive effort to characterize morphology, cell cycle (revealing six distinct stages) and nucleoid dynamics of *D. radiodurans*. Its importance is in the fact that such characterization is rare for coccoid bacteria compared to rod- or crescent shaped model bacteria. The manuscript is very clearly written and overall a pleasure to read. The data presented are convincing.

There is a small number of points – summarized below - that may need attention to further improve the manuscript:

1. Throughout the manuscript the authors use the terms 'single-molecule and ensemble studies' when referring to 'single-molecule tracking' and 'FRAP analysis'. It may be helpful for the reader to explicitly point out that these are also microscopy studies, as elsewhere in the paper. Otherwise the terms used might equally well refer to biochemical *in vitro* approaches.

2. P. 3/p. 17 the authors state the observed nucleoid dynamics in terms of diverse cell cycle dependent morphology is facilitated by dynamic binding of the histone-like HU protein. Whereas observations are supported by the data it is not obvious that the two are directly connected. This needs rewording.

Note that indeed very little information on *in vivo* binding of histone-like proteins is available apart from that reported for *E. coli* H-NS in reference 37. Also *in vitro* not much information is available, but note that Dame and colleagues published off-rates for H-NS and HU *in vitro* (Dame et al., 2006; Dame et al., 2013). According to those studies both proteins have high off-rates when individual binding domains are studied, but cooperativity ensure that H-NS filaments are quite stable, while also for HU high DNA binding density effectively stabilizes the complex.

3. P. 3: When referring to the binding of the HU protein the authors use the term 'coating'. This implies dense coverage, while to my knowledge no such information is available.

4. P. 8: For staining of the nucleoid the authors employ both the Syto9 DNA binding dye and genomically encoded HU-mCherry. Figure S7 shows global colocalization of the two types of labels. Again (see comment 3) the authors use the term 'coating' of DNA while the microscopy images do not show anything beyond a global binding – binding density along the genome can not be extracted from these images.

5. Figure 3: the authors state that for both types of labelling (see comment 4) the 'same nucleoid structures' are observed. However, in figure 3 they only show the data for Syto9 labelling. Their statement would be better supported if they would also include the same analysis for the HU-mCherry labelled cells.

6. P.11/figure S7E: The authors compare the volume of the nucleoid in wt and HU-mCherry expressing cells by Syto9-staining and note that the volume in the HU-mCherry encoding strain is higher. This is attributed to 'steric hindrance by the mCherry fusion protein'. This could be worded more explicitly, saying for instance that the 'mCherry label may interfere with the DNA binding properties of HU' (assuming the amounts of fusion and non-fusion protein expressed are the same). It will also be fair to note this in the context of the measurements of the effective diffusion coefficients, where due to a PAmCherry effect somewhat higher values are possibly obtained than for the wt protein.

7. P. 13/figure 5c: the authors make a point of the mobile fraction of protein in their FRAP experiments being 95%. It is not directly obvious that the starting intensity level and final level

are different. Please substantiate. If the difference is significant, what could be the reason for 5% of the protein having lower dissociation rate?

8. P. 16: In the discussion the authors say they identified 8 distinct phases. Please clarify as figure 1 only depicts 6 phases.

Reviewer #2 (Remarks to the Author):

This is a thorough and very well written manuscript describing the morphological changes that occur during the cell cycle of *Deinococcus radiodurans*, as well as the dynamics of the nucleoid. The work is descriptive but novel, giving detailed view of the cell cycle of this interesting bacterium. I was very impressed by the quality and level of detail of the presented data.

General comments

Line 177 – Why is BADA not incorporated in the septum? This is different from most bacteria.

Line 178 – Nile Red labels the membrane, so it would be more accurate to say “and thus labels the whole cell membrane including in newly synthesized septa”

Line 331 – Is it known if the number of copies of the genome is lower in stationary phase?

Line 357 – Figure S8 is missing

Line 458 – “progressive increase in the curvature of the central septum” – what do the authors mean by “curvature”? The central septum is not a flat disk?

Line 603 – Please define “saturated pre-culture”. Also, chloramphenicol causes nucleoid condensation. Were these experiments also done in the absence of the antibiotic?

Figure 1 – Phases 2, 3 and 4 are defined qualitatively. Given that authors have measured the length of the growing septum in each cell, definition of phases could be done in a more objective way, for example by determining the fraction of the septum that is made for transition between each of these phases.

Figure 3 – Although this is a small detail, it would be good to represent toroids and squares with a similar colour (eg light and dark tone) as they correspond to very similar (or identical) organization of the nucleoid.

Figure 6, panel B – state that number of foci comes from 3D microscopy data. This is stated in the methods, but it is relevant information given that the authors see less foci than a previous study, which could be due to lack of visualization of out of focus foci, if the data resulted from 2D images. Also, while it is clear from the images shown that the ter is sometimes in the center of the toroid, it is not so clear that the oriCs have a radial distribution. Perhaps the authors can show some enlarged images showing more clearly this distribution?

Figure S1 – Line 5 of legend – “The same three populations ...” or the six populations?

Figure S2 – Mentioning that Scentral corresponds to “length of the old division septum originating from the previous cell cycle” may be misleading, as the old septum was probably longer, given that splitting of the daughter cells during the cell cycle will reduce the length of the plane of attachment between daughter cells. It may be more clear to call this “septum” by a different name, for example “attachment plate between daughter cells”. This would

also solve the problem mentioned in FigS3 of phase 6 cells having two central septa: the old septum originating from the previous cell cycle (attachment plate) and the new one that has just closed.

Fig S3 – This figure has a lot of information. However, it may not be easy for the reader to extract conclusions for the presentation of all the measurements.

For example, panel D gives information about the invagination leading to the splitting of daughter cells. This could be clearly stated.

What does panel F inform about? One could assume that the fact that there is no statistically significant difference between the inner and outer septum indicates that synthesis of the septum is synchronous from both sides, which is not the case. Would it be possible to plot the correlation of size of inner and outer septum for each cell and use that as a measure of synchronicity?

And what about panel C? What information regarding the cell cycle progression do the authors want the reader to take from these data?

The relevance/significance of each panel should be explained.

Fig S4, panel D – It would be more informative to split the data per phase of the cell cycle (6, 7 and 8), similarly to other figures.

Fig S5 – Panel B is not needed in this figure. As far as I could understand, there is no relation between panels A and B. How was the time in panel A defined? I assume cells are not synchronized, so time 0 should be the time of start of the cell cycle for each analysed cell. It does not correspond to times of growth phase, correct?

Figure S7 – data for *E. coli* is not very informative because it is not given as a function of the cell cycle. There is a large dispersion (from 0.4-0.95) in the fraction of the cell volume occupied by the nucleoid. Would this be lower if analysis was done for different phases of the cells cycle?

Figure S8 is missing (mentioned in line 357)

Minor comments

Line 42 – brackets missing after reference 10

Line 48- "and" after "oriC" should not be italicized

Line 63 - remove "copies" after "4 to 10"

Line 195 – reference in brackets should be replaced by number

Figure 5, title – Dynamics without capital letter

Figure S2, panel D – X3-X6 should be indicated below the graphs (in the x axis) similarly to what was done in panel C. There are two symbols that seem cut above the graph on the right. Also, in the legend of panel D indicate which graph corresponds to length and width (by mentioned right and left for example).

Reviewer #3 (Remarks to the Author):

Deinococcus radiodurans is a species that has captured the imagination since its initial description. It was the first organism identified as being extremely resistant to ionizing radiation, and since that discovery developing an explanation for the species' origin and identifying the mechanisms employed to achieve radioresistance have been the focus of almost all serious study of this species. Unfortunately, in the push to explain radioresistance other remarkable characteristics associated with the species have not been investigated. In this study, Dr. Timmins and her colleagues examine the unusual manner in which *D. radiodurans* undergoes cell division by applying sophisticated microscopic techniques to follow changes that occur as the species moves

through its cell cycle.

The work is a tour de force. The authors document with notable clarity coordinated morphological changes in the cell and nucleoid as the species progresses through the cell cycle, illustrating that *D. radiodurans* nucleoids reproducibly adopt multiple distinct configurations that correlate with distinct phases of the cell cycle. While prior studies over the past 40 years have hinted at the complexity of cell division in this species, this is the only work to follow those changes as a function of time. That capability has allowed the authors to illustrate how plastic the nucleoid is in this species, and how precisely the events of cell division are orchestrated. The work confirms earlier reports that the nucleoid is condensed and that newly forming septa grow toward each other as if they were a pair of curtains closing, leaving an ever-narrowing gap the width of the cell as septation proceeds. To my knowledge, these phenomena have not been examined in this detail in any other species previously. In addition, the work indicates that *oriC* loci of chromosomes are radially distributed around clustered *ter* sites maintained at the center of cells, a novel arrangement that may be applicable to nucleoid structure in other cocci. The large size of *D. radiodurans* makes such measurements feasible and suggests that this species may function as an effective model for the study of nucleoid dynamics in cocci in the future.

I have no issues with the manuscript. The work is carefully done, at least two complementary approaches are taken for each study, and the results obtained are in agreement, multiple measurements are recorded and appropriate statistical analyses performed.

Reviewers' comments:

Reviewer #1 (Remarks to the Author):

The authors do an impressive effort to characterize morphology, cell cycle (revealing six distinct stages) and nucleoid dynamics of *D. radiodurans*. Its importance is in the fact that such characterization is rare for coccoid bacteria compared to rod- or crescent shaped model bacteria. The manuscript is very clearly written and overall a pleasure to read. The data presented are convincing.

There is a small number of points – summarized below - that may need attention to further improve the manuscript:

1. Throughout the manuscript the authors use the terms ‘single-molecule and ensemble studies’ when referring to ‘single-molecule tracking’ and ‘FRAP analysis’. It may be helpful for the reader to explicitly point out that these are also microscopy studies, as elsewhere in the paper. Otherwise the terms used might equally well refer to biochemical in vitro approaches.

➤ This has been modified.

2. P. 3/p. 17 the authors state the observed nucleoid dynamics in terms of diverse cell cycle dependent morphology is facilitated by dynamic binding of the histone-like HU protein. Whereas observations are supported by the data it is not obvious that the two are directly connected. This needs rewording.

➤ This has been reworded so as to make clear that this is a hypothesis and not a firm statement.

Note that indeed very little information on in vivo binding of histone-like proteins is available apart from that reported for *E. coli* H-NS in reference 37. Also in vitro not much information is available, but note that Dame and colleagues published off-rates for H-NS and HU in vitro (Dame et al., 2006; Dame et al., 2013). According to those studies both proteins have high off-rates when individual binding domains are studied, but cooperativity ensure that H-NS filaments are quite stable, while also for HU high DNA binding density effectively stabilizes the complex.

3. P. 3: When referring to the binding of the HU protein the authors use the term ‘coating’. This implies dense coverage, while to my knowledge no such information is available.

➤ In *D. radiodurans*, HU has been reported to be one of the most abundant proteins in the cell and our finding that HU-mCherry colocalizes with the Syto9 stained genomic DNA all over the nucleoid, suggests that HU most likely ‘coats’ entirely the DNA, although of course we do not have any data allowing us to directly evaluate the density of HU. The word ‘coats’ has been replaced by ‘associates’ or ‘colocalizes’.

4. P. 8: For staining of the nucleoid the authors employ both the Syto9 DNA binding dye and genomically encoded HU-mCherry. Figure S7 shows global colocalization of the two types of labels. Again (see comment 3) the authors use the term ‘coating’ of DNA while the microscopy images do not show anything beyond a global binding – binding density along the genome can not be extracted from these images.

➤ Please see response to point 3.

5. Figure 3: the authors state that for both types of labelling (see comment 4) the ‘same nucleoid structures’ are observed. However, in figure 3 they only show the data for Syto9 labelling. Their statement would be better supported if they would also include the same analysis for the HU-mCherry labelled cells.

➤ As suggested by the reviewer, an extra panel has now been added to Fig. S8 (previously Fig. S7) to illustrate the different nucleoid morphologies observed with HU-mCherry. This new array allows a direct comparison with the nucleoid morphologies observed in Syto9-stained cells. Because of the reduced photostability of mCherry compared to Syto9, performing 3D-timelapse imaging on HU-mCherry labelled cells was more challenging and was thus less suitable for an in-depth study of the nucleoid morphology and volume as a function of the cell cycle. This is why we present a full analysis of the Syto9 stained cells, together with colocalization experiments to illustrate that very similar staining and nucleoid shapes were observed using both staining strategies.

6. P.11/figure S7E: The authors compare the volume of the nucleoid in wt and HU-mCherry expressing cells by Syto9-staining and note that the volume in the HU-mCherry encoding strain is higher. This is attributed to ‘steric hindrance by the mCherry fusion protein’. This could be worded more explicitly, saying for instance that the ‘mCherry label may interfere with the DNA binding properties of HU’ (assuming the amounts of fusion and non-fusion protein expressed are the same. It will also be fair to note this in the context of the measurements of the effective diffusion coefficients, where due to a PAmCherry effect somewhat higher values are possibly obtained than for the wt protein.

➤ This has been clarified according to the reviewer’s suggestion.

7. P. 13/figure 5c: the authors make a point of the mobile fraction of protein in their FRAP experiments being 95%. It is not directly obvious that the starting intensity level and final level are different. Please substantiate. If the difference is significant, what could be the reason for 5% of the protein having lower dissociation rate?

➤ The mobile fraction of HU-mCherry determined in our FRAP experiments is very close to 100%, and when bleaching a significant (in our case 60% of the total fluorescence) fraction of the fluorescence signal, full recovery (100% of pre-bleach state) is not expected even with a uniform population. The value extracted from our fitted curve was indeed 95% and the 5% difference was most likely within the error of the measurements. To verify this, we have included more replicates to make the data more robust and reprocessed our data to try to minimize possible errors: in particular we have finely corrected the xy drift in our images and have selected the normalized datasets (N=10) that could be suitably fitted ($R^2 > 0.7$) to a single term equation. The estimated mobile fraction and half-life derived from these newly processed data and presented in Fig. 5d are respectively of 99% (+/- 1%) and 1.03 sec. Our data thus clearly suggest that a vast majority of the HU-mCherry is mobile and the remaining fraction is not statistically significant. A more detailed description of the FRAP data analysis has been added to the Methods section and the newly processed data is presented in Fig. 5d. The raw and normalized FRAP data are also provided in the Source Data file.

8. P. 16: In the discussion the authors say they identified 8 distinct phases. Please clarify as figure 1 only depicts 6 phases.

- The 8 phases mentioned in the discussion correspond to the 6 phases observed in exponentially growing cells (and illustrated in Fig. 1) plus the two additional phases observed in stationary cells (and shown in Fig. S4).

Reviewer #2 (Remarks to the Author):

This is a thorough and very well written manuscript describing the morphological changes that occur during the cell cycle of *Deinococcus radiodurans*, as well as the dynamics of the nucleoid. The work is descriptive but novel, giving detailed view of the cell cycle of this interesting bacterium. I was very impressed by the quality and level of detail of the presented data.

General comments

Line 177 – Why is BADA not incorporated in the septum? This is different from most bacteria.

- BADA and HADA were in fact incorporated in the septum immediately after the staining, but this incorporation was short-lived (<10 min), perhaps as a result of the processing and remodeling of the peptidoglycan layer within the growing septal regions. This is why for the cell wall experiments (Fig. 2) the cells were rinsed to remove unbound dyes and imaged after a short recovery time (15min) allowing for the stable incorporation of the BADA/HADA dyes into the peripheral cell walls only.

Line 178 – Nile Red labels the membrane, so it would be more accurate to say “and thus labels the whole cell membrane including in newly synthesized septa”

- This has been modified according to the reviewer’s suggestion.

Line 331 – Is it know if the number of copies of the genome is lower in stationary phase?

- An early study by Hansen et al (J. Bact. 1978) suggested that *D. radiodurans* still possessed several genome copies in stationary phase. Recent deep sequencing data obtained by P. Servant and colleagues (manuscript in preparation) confirm this and indicate that the number of genome copies remains approximately the same (between 5 and 8) in both stationary and exponentially growing cells as shown in the figure below extracted from the manuscript in preparation:

Figure legend: Absolute quantification per cell based on deep sequencing data normalized by qPCR based absolute quantification of oriC1. Values of C1 and C2 from regression curve; values of MP1 and CP1 from density of 2000bp around position 0. C1: chromosome 1; C2: chromosome 2; MP1: Megaplasmid 1; CP1: plasmid 1)

Line 357 – Figure S8 is missing

- The reference to Figure S8 has been removed. The data is presented in Fig. 5b.

Line 458 – “progressive increase in the curvature of the central septum” – what do the authors mean by “curvature”? The central septum is not a flat disk?

- This sentence has been modified. We meant that the curvature of the cell walls of the two cells progressively increases as the cells progress through the cell cycle, thereby leading to a reduced length of the flat, shared central septum.

Line 603 – Please define “saturated pre-culture”. Also, chloramphenicol causes nucleoid condensation. Were these experiments also done in the absence of the antibiotic?

- The term ‘saturated pre-culture’ has been corrected. These were cultures grown overnight prior to dilution. In the case of *ori* and *ter* labelled strains (GY15787 and GY15800), chloramphenicol was added to the cultures at a concentration of 3.4 µg/mL to maintain the plasmid expressing ParB-GFP. The HU-mCherry nucleoids in these strains were indistinguishable from those observed in the GY15743 strain (HU-mCherry only) grown in the absence of chloramphenicol, suggesting no significant nucleoid condensation.

Figure 1 – Phases 2, 3 and 4 are defined qualitatively. Given that authors have measured the length of the growing septum in each cell, definition of phases could be done in a more objective way, for example by determining the fraction of the septum that is made for transition between each of these phases.

- The 6 phases have indeed been defined qualitatively. Phase 1 and 2 cells can be distinguished by their ellipticity and invagination, while phase 3 cells possess foci corresponding to the start of the growing septa. Such features can readily be distinguished and do not require precise measurements. Distinguishing between phase 4 and 5 cells is indeed more ambiguous, since they both correspond to cells with growing septa and we chose to simply define phase 5 cells as those in which septal closure was almost complete. Phase 4 cells are thus more diverse, corresponding to cells with growing septa at different stages. We agree with the reviewer that defining the fraction of the septum that has been formed would make this definition more objective. This has been done in the revised manuscript and Fig. 1 and S3 have been updated accordingly. Phase 5 cells have now been defined as cells with growing septa in which the sum of the lengths of the interior and exterior septa is at least 80% of the distance P (see Fig. S2, where P=distance between central septum and the opposite side of the cell). As can be seen in the new figures, this new definition does not change our conclusions in any way, and only 4-5 cells were exchanged between

Phases 4 and 5. We thank the reviewer for this suggestion that has helped to make our classification more robust and unbiased.

Figure 3 – Although this is a small detail, it would be good to represent toroids and squares with a similar colour (eg light and dark tone) as they correspond to very similar (or identical) organization of the nucleoid.

- The colour of the squares in the graph presented in Fig. 3 has been changed to dark blue and the non-defined nucleoids are now in pink.

Figure 6, panel B – state that number of foci comes from 3D microscopy data. This is stated in the methods, but it is relevant information given that the authors see less foci than a previous study, which could be due to lack of visualization of out of focus foci, if the data resulted from 2D images. Also, while it is clear from the images shown that the *ter* is sometimes in the center of the toroid, it is not so clear that the *oriC*s have a radial distribution. Perhaps the authors can show some enlarged images showing more clearly this distribution?

- The figure legend has been modified to specify that the numbers were retrieved from 3D images. It is not obvious to see the radial distribution of *OriC* in individual images, since there are only a few foci in each cell, and it is only when we compare the distribution of *ter* and *oriC* foci in a large number of cells that this becomes clearly apparent. Fig. 6d has been nonetheless modified to present close-up views of both *ter* and *oriC* labelled cells.

Figure S1 – Line 5 of legend – “The same three populations ...” or the six populations?

- Here we are referring to the three independent experiments/cultures used to perform these measurements and not to the 6 different phases.

Figure S2 – Mentioning that *Scentral* corresponds to “length of the old division septum originating from the previous cell cycle” may be misleading, as the old septum was probably longer, given that splitting of the daughter cells during the cell cycle will reduce the length of the plane of attachment between daughter cells. It may be more clear to call this “septum” by a different name, for example “attachment plate between daughter cells”. This would also solve the problem mentioned in FigS3 of phase 6 cells having two central septa: the old septum originating from the previous cell cycle (attachment plate) and the new one that has just closed.

- Following the reviewer’s recommendation we have changed the nomenclature describing the old and new septa. We have now defined the central septum originating from the previous cell cycle as S_{-1} and the newly growing septum as S_0 to avoid any confusion. This has been modified accordingly in the main text and figure legends.

Fig S3 – This figure has a lot of information. However, it may not be easy for the reader to extract conclusions for the presentation of all the measurements.

For example, panel D gives information about the invagination leading to the splitting of daughter cells. This could be clearly stated.

What does panel F inform about? One could assume that the fact that there is no statistically significant difference between the inner and outer septum indicates that synthesis of the septum is synchronous from both sides, which is not the case. Would it be possible to plot the correlation of size of inner and outer septum for each cell and use that as a measure of synchronicity?

And what about panel C? What information regarding the cell cycle progression do the authors want the reader to take from these data?

The relevance/significance of each panel should be explained.

- Figure S3 has now been reorganized and the relevance and significance of each panel has been clarified in the figure legend.
In this new Fig. S3, panel A presents the different measurements used for determining cell size based on the use of fitted ellipses, and the extent of ellipticity of the cells as a function of the cell phase. The graph presenting the distance to the central septum (panel C in the original version of Fig. S3) has been removed. Although these measurements were essential for determining the volumes of the cell as illustrated in Fig. S2, they are not critical for the understanding of the paper and do not add any new information compared to the cell volume graph presented in Fig. 1D.
Panel B now presents the lengths of the septa originating from the previous cell division (now named S_{-1}) as a function of the cell cycle, which indeed is an indirect measure of the extent of invagination.
Panel C presents the lengths of the two septal fragments composing the S_0 septa in phases 4 and 5. This data shows that septal growth occurs equally from both sides since no significant difference in length is observed between S_{ext} and S_{int} for each phase. This data also highlights the substantially longer S_0 septa in phase 5 vs phase 4 cells (criterion used to distinguish these two phases; see response above). However, these data do not inform us about the synchronicity or speed of septal growth from both sides and this has been corrected in the manuscript (2nd paragraph of the discussion: "...no difference in growth speed was observed..." has been corrected to "...no difference in length was observed..."). Appropriate references to this new figure have been added at relevant places in the manuscript.

Fig S4, panel D – It would be more informative to split the data per phase of the cell cycle (6, 7 and 8), similarly to other figures.

- The analysis of stationary phase cells was only performed on phase 6 cells so as to allow us to compare these measurements with those obtained in exponentially growing phase 6 cells. A more extensive study of stationary cells and nucleoids will be the focus of a future study.

Fig S5 – Panel B is not needed in this figure. As far as I could understand, there is no relation between panels A and B. How was the time in panel A defined? I assume cells are not synchronized, so time 0 should be the time of start of the cell cycle for each analysed cell. It does not correspond to times of growth phase, correct?

- The reviewer is correct, there is no relation between panels A and B. Panel B has therefore been transferred to a separate figure (now Fig. S6) for clarity. In panel A, the

cells are indeed not synchronized, and we simply defined the start of the cell cycle (beginning of phase 1) as time 0 and the other data points correspond to the start of the subsequent phases based on the average durations of the different phases determined in our timelapse experiments (Fig. 1e). This has been clarified in the figure legend.

Figure S7 – data for *E. coli* is not very informative because it is not given as a function of the cell cycle. There is a large dispersion (from 0.4-0.95) in the fraction of the cell volume occupied by the nucleoid. Would this be lower if analysis was done for different phases of the cells cycle?

- In this work, the data for *E. coli* allows us to compare the average cell and nucleoid volumes of *E. coli* vs. *D. radiodurans* and was also used as a reference for validating our measurements and calculations of cell and nucleoid sizes. So we feel that the presented data is informative and has been maintained (now in Fig. S9 of the revised manuscript). We agree that it would be interesting to perform a more in-depth analysis of *E. coli* cell and nucleoid dimensions at different phases of the cell cycle, but this represents a major task and defining different phases of the *E. coli* cell cycle would be challenging. The large dispersion in the fraction of the cell volume occupied by the nucleoid is mostly a consequence of the large variation in size of the *E. coli* cells, but the average fraction is very similar to that reported in earlier studies. We were particularly interested in the level of compaction of the nucleoid and the fraction of the cell volume that it occupies in *D. radiodurans* vs. *E. coli*, and our data in *D. radiodurans* show that although we do see some variation during the cell cycle, this is only minor compared to the major difference in cell fraction observed between the two bacterial species. We therefore believe that such an in-depth study of *E. coli* cells would not modify the conclusions of this present study.

Figure S8 is missing (mentioned in line 357)

- Reference to figure S8 has been removed and instead reference to Fig. 5b has been added.

Minor comments

Line 42 – brackets missing after reference 10

Line 48- “and” after “oriC” should not be italicized

Line 63 - remove “copies” after “4 to 10”

Line 195 – reference in brackets should be replaced by number

Figure 5, title – Dynamics without capital letter

- We have addressed all of these minor comments.

Figure S2, panel D – X3-X6 should be indicated below the graphs (in the x axis) similarly to what was done in panel C. There are two symbols that seem cut above the graph on the right. Also, in the legend of panel D indicate which graph corresponds to length and width (by mentioned right and left for example).

➤ Figure S2 has been modified accordingly.

Reviewer #3 (Remarks to the Author):

Deinococcus radiodurans is a species that has captured the imagination since its initial description. It was the first organism identified as being extremely resistant to ionizing radiation, and since that discovery developing an explanation for the species' origin and identifying the mechanisms employed to achieve radioresistance have been the focus of almost all serious study of this species. Unfortunately, in the push to explain radioresistance other remarkable characteristics associated with the species have not been investigated. In this study, Dr. Timmins and her colleagues examine the unusual manner in which *D. radiodurans* undergoes cell division by applying sophisticated microscopic techniques to follow changes that occur as the species moves through its cell cycle.

The work is a tour de force. The authors document with notable clarity coordinated morphological changes in the cell and nucleoid as the species progresses through the cell cycle, illustrating that *D. radiodurans* nucleoids reproducibly adopt multiple distinct configurations that correlate with distinct phases of the cell cycle. While prior studies over the past 40 years have hinted at the complexity of cell division in this species, this is the only work to follow those changes as a function of time. That capability has allowed the authors to illustrate how plastic the nucleoid is in this species, and how precisely the events of cell division are orchestrated. The work confirms earlier reports that the nucleoid is condensed and that newly forming septa grow toward each other as if they were a pair of curtains closing, leaving an ever-narrowing gap the width of the cell as septation proceeds. To my knowledge, these phenomena have not been examined in this detail in any other species previously. In addition, the work indicates that *oriC* loci of chromosomes are radially distributed around clustered *ter* sites maintained at the center of cells, a novel arrangement that may be applicable to nucleoid structure in other cocci. The large size of *D. radiodurans* makes such measurements feasible and suggests that this species may function as an effective model for the study of nucleoid dynamics in cocci in the future.

I have no issues with the manuscript. The work is carefully done, at least two complementary approaches are taken for each study, and the results obtained are in agreement, multiple measurements are recorded and appropriate statistical analyses performed.